# Hinge and Transmembrane Domains of Chimeric Antigen Receptor Regulate Receptor Expression and Signaling Threshold

**DOI:** 10.3390/cells9051182

**Published:** 2020-05-09

**Authors:** Kento Fujiwara, Ayaka Tsunei, Hotaka Kusabuka, Erika Ogaki, Masashi Tachibana, Naoki Okada

**Affiliations:** Project for Vaccine and Immune Regulation, Graduate School of Pharmaceutical Sciences, Osaka University, Osaka 565-0871, Japan; fujiwara-k@phs.osaka-u.ac.jp (K.F.); papaian0720@gmail.com (A.T.); yakuzai.gallios@gmail.com (H.K.); jyugonkun@yahoo.co.jp (E.O.); tacci@phs.osaka-u.ac.jp (M.T.)

**Keywords:** chimeric antigen receptor, CAR-T cell therapy, structure-activity relationship, hinge domain, transmembrane domain

## Abstract

Chimeric antigen receptor (CAR)-T cells have demonstrated significant clinical potential; however, their strong antitumor activity may cause severe adverse effects. To ensure efficacy and safe CAR-T cell therapy, it is important to understand CAR’s structure–activity relationship. To clarify the role of hinge and transmembrane domains in CAR and CAR-T cell function, we generated different chimeras and analyzed their expression levels and antigen-specific activity on CAR-T cells. First, we created a basic CAR with hinge, transmembrane, and signal transduction domains derived from CD3ζ, then we generated six CAR variants whose hinge or hinge/transmembrane domains originated from CD4, CD8α, and CD28. CAR expression level and stability on the T cell were greatly affected by transmembrane rather than hinge domain. Antigen-specific functions of most CAR-T cells depended on their CAR expression levels. However, CARs with a CD8α- or CD28-derived hinge domain showed significant differences in CAR-T cell function, despite their equal expression levels. These results suggest that CAR signaling intensity into T cells was affected not only by CAR expression level, but also by the hinge domain. Our discoveries indicate that the hinge domain regulates the CAR signaling threshold and the transmembrane domain regulates the amount of CAR signaling via control of CAR expression level.

## 1. Introduction

Adoptive T-cell transfer has been heralded as an ideal cancer treatment strategy that causes effective regression of primary cancer, suppresses metastasis and recurrence, and has few side effects on normal tissues [1,2]. However, clinical applications have been limited by the difficulty of preparing tumor-specific cytotoxic T cells of sufficient quality and quantity for treatment as well as inadequate accumulation of administered T cells to the tumor site. The creation of T cells that genetically express a chimeric antigen receptor (CAR) and their use in next-generation adoptive T-cell transfer have been proposed to overcome current treatment limitations [3,4,5].

The success of anti-CD19 CAR-T cell therapy for patients with B-cell lymphoma has proven that T cell genetic modification enables the rapid preparation of tumor-specific T cells with long-term antitumor effects [6,7,8,9]. The administered CAR-T cells not only exert cytotoxic activity by recognizing the target antigen on tumor cells via CAR, but also proliferate and survive for a long time in vivo; therefore, even a single dose can continue to elicit sustained antitumor effects [8,9]. While the effectiveness and safety of CAR-T cell therapy has been confirmed by several studies, it is based on individually designed and constructed CARs. This means that there are few structural approaches aimed at optimizing CAR function, and a system that systematically analyzes the structure–activity relationship of CAR has not been constructed yet.

CAR is an artificial transmembrane receptor that connects the extracellular antigen recognition domain (ARD), hinge domain (HD), transmembrane domain (TMD), and intracellular signal transduction domain (STD) in tandem. All T cells collected from cancer patients acquire antigen specificity by expressing CAR [3,4]. Although it is not difficult to imagine that structural modification of CAR components will be a factor that defines CAR functionality such as “affinity and specificity to target molecule”, “signal transmission efficiency and intensity”, and “signal type”, the structure–activity relationship of CAR has not been systematically verified. In addition, CAR-T cell therapy has been associated with the onset of side effects such as on-target off-tumor toxicity [10] and cytokine release syndrome [6,7]. Therefore, it is paramount to establish a design and tuning methodology that can optimize CAR-T cell function as well as to select the appropriate CAR target molecule.

We hypothesized that additional evidence on the relationship between CAR structure and CAR-T cell function could introduce structural information into CAR design, thus increasing therapeutic efficacy while reducing side effects. To clarify the role of HD/TMD in CAR activity and CAR-T cell function, we created various HD-modified and HD/TMD-modified CARs, and systematically compared and analyzed their expression levels as well as the antigen-specific functional strength of T cells into which those genes or mRNAs were introduced.

## 2. Materials and Methods

### 2.1. Cell Lines

Human Plat-E cells were obtained from Cell Biolabs (San Diego, CA) and cultured in Dulbecco’s modified Eagle medium (DMEM) supplemented with 10% fetal bovine serum (FBS), 1 μg/mL puromycin, and 10 μg/mL blasticidin. Murine EL4 cells were obtained from Cell Resource Center for Biomedical Research, Institute of Development, Aging, and Cancer, Tohoku University (Sendai, Japan), and cultured in RPMI 1640 medium supplemented with 10% FBS and 50 μM 2-mercaptoethanol. Murine L1.2 cells were kindly provided by Prof. Takashi Nakayama (Kindai University Faculty of Pharmacy, Higashi-osaka, Japan) and cultured in RPMI 1640 medium supplemented with 10% FBS. Murine vascular endothelial growth factor receptor 2 (VEGFR2)-expressing EL4 (mVEGFR2^+^ EL4) cells and human VEGFR2-expressing L1.2 (hVEGFR2^+^ L1.2) cells were generated by transducing the respective cell lines with a retroviral vector (Rv) containing the mVEGFR2 or hVEGFR2 gene and the puromycin resistance cassette. The cells were grown in the culture medium of each untransduced cell line supplemented with puromycin. All cells were maintained in a humidified atmosphere of 5% CO_2_ at 37 °C.

### 2.2. Mice

Female C57BL/6 mice (6–8-weeks old) were purchased from SLC (Hamamatsu, Japan) and were maintained in the experimental animal facility at Osaka University. Care and use of laboratory animals complied with the guidelines and policies of the Act on Welfare and Management of Animals in Japan. Protocols and procedures were approved by the Animal Care and Use Committee of Osaka University.

### 2.3. Construction of CAR Structural Variants

The mVEGFR2-specific basic CAR construct contained an Igκ-chain leader sequence, HA-tag, anti-mVEGFR2 single-chain variable fragment (scFv; clone avas12) [11,12], and murine CD3ζ-derived HD/TMD/STD (Figure 1A). The construct was subcloned into pMXs-Puro (Cell Biolabs). To construct HD-modified and HD/TMD-modified CARs, HD/TMD fragments with various combinations of murine CD3ζ, CD4, CD8α, and CD28 were synthesized by polymerase chain reaction amplification. Each HD/TMD fragment and murine CD3ζ-STD fragment was assembled by Gibson assembly to include a *Sac*II site at the 5′ end and a *Not*I site at the 3′ end. These synthetic fragments were digested with *Sac*II and *Not*I restriction enzymes (New England Biolabs, Ipswich, MA) and individually ligated into similarly digested pMXs-Puro/CAR. The generated mVEGFR2-specific CARs are denoted as mV/HD/TMD/STD, with the basic one being mV/3z/3z/3z.

The hVEGFR2-specific CARs contained an Igκ-chain leader sequence, HA-tag, anti-hVEGFR2 scFv [13,14], human CD8α- or human CD28-HD/TMD, human CD28-STD, and human CD3ζ-STD (Figure 1B). Each construct was subcloned into pcDNA3.1-Zeo (Thermo Fisher Scientific, Waltham, MA). CAR with human CD8α-HD/TMD is denoted as hV/h8a/h8a/h28-h3z and CAR with human CD28-HD/TMD is denoted as hV/h28/h28/h28-h3z.

Sequence integrity of all plasmids was confirmed by DNA sequencing (Fasmac Co., Atsugi, Japan). The amino acid (AA) sequences of various immune molecules used in this study are summarized in Figure 1C.

### 2.4. Production of CAR-T Cells

Murine CAR-T cells were produced as previously described [15]. Briefly, the Rv packaging CAR gene was produced by transfecting Plat-E cells with pMXs-Puro/CAR. Murine T cells were activated by using anti-CD3ε mAb (clone 145-2C11, Bioxcell, West Lebanon, NH) and an anti-CD28 mAb (clone 37.51, Bioxcell), and then transduced with Rv-bound Retronectin (Takara Bio, Kusatsu, Japan) under anti-CD3ε/CD28 mAbs stimulation. After Rv-transduction, CAR-T cells were cultured in RPMI 1640 medium supplemented with 10% FBS, 50 μM 2-mercaptoethanol, MEM Non-essential Amino Acids Solution (FUJIFILM Wako Pure Chemical, Osaka, Japan), 10 U/mL interleukin-2 (Peprotech, Rocky Hill, NJ), and 5 μg/mL puromycin.

Human CAR-T cells were produced as previously described [13]. Briefly, CAR mRNA was transcribed using the mMESSAGE mMACHINE T7 Ultra kit (Thermo Fisher Scientific) and linearized pcDNA3.1-Zeo/CAR. Human T cells were activated by using an anti-CD3 mAb (Janssen Pharmaceutica, Beerse, Belgium) for 13 days, and CAR mRNA was introduced by electroporation.

### 2.5. Flow Cytometry Analysis for Surface Expression and Antigen-Binding Capacity of CAR

Murine CAR-T cells were incubated with an anti-mouse CD16/CD32 mAb (clone 93, Biolegend, San Diego, CA), and then stained with the Zombie Aqua Fixable Viability Kit (Biolegend) and PE-Cy7-labeled anti-CD8α mAb (clone 53-6.7, Biolegend). CAR expression was evaluated using APC-labeled anti-HA.11 Epitope Tag mAb (clone 16B12, Biolegend) or APC-labeled mouse IgG1 isotype control mAb (clone MOPC-21, Biolegend). The mVEGFR2-binding of CAR was evaluated using recombinant Mouse VEGFR2/Flk-1 Fc Chimera (mVEGFR2-Fc, R&D Systems, Minneapolis, MN) and Alexa Fluor 647-labeled anti-His-tag mAb (clone OGHis, Medical & Biological Laboratories, Nagoya, Japan).

Human CAR-T cells were incubated with Human TruStain FcX (Biolegend) and then stained with the LIVE/DEAD Fixable Aqua Dead Cell Stain Kit, PE-Cy7-labeled anti-CD8α mAb (clone HIT8α, Biolegend), and APC-labeled anti-HA mAb or APC-labeled mouse IgG1 isotype control mAb.

Immunofluorescence was measured using a BD FACS Canto II (BD Biosciences, Franklin Lakes, NJ), and data were analyzed using FlowJo software (FlowJo LLC, Ashland, OR). Surface expression level or mVEGFR2-binding of CAR were represented by the geometric mean fluorescence intensity (GMFI) ratio, calculated according to the following formula: GMFI ratio = (GMFI of CAR-T cells stained with anti-HA mAb or anti-His-tag mAb with mVEGFR2-Fc)/(GMFI of CAR-T cells stained with mouse IgG1 isotype control mAb or anti-His-tag mAb without mVEGFR2-Fc). 

### 2.6. Reverse Transcription and Quantitative Polymerase Chain Reaction

Total RNA was isolated from CAR-T cells with TRIzol reagent (Thermo Fisher Scientific), and reverse-transcribed using Super Script III Reverse Transcriptase (Thermo Fisher Scientific). CAR cDNA was detected with the Custom TaqMan Gene Expression Assay (Thermo Fisher Scientific) for anti-mVEGFR2 scFv or anti-hVEGFR2 scFv. CAR and GAPDH expression levels were measured using the CFX96 Real-Time PCR Detection System (Bio-Rad Laboratories, Hercules, CA).

### 2.7. Western Blot Analysis

CAR-T cells on day 0 were lysed with cell lysis buffer containing 50 mM Tris-HCl pH 7.4, 150 mM NaCl, 1 mM EDTA-Na_2_, 5% glycerol, 1% Triton-X, and protease inhibitor cocktail (Nacalai Tesque, Kyoto, Japan). The supernatant was collected as the cell lysate sample, its protein content was quantified by the DC protein assay (Bio-Rad Laboratories). Lysate sample was mixed with Laemmli sample buffer (Bio-Rad Laboratories) containing 20% 2-mercaptoethanol or not, and heat denatured. Next, lysate samples were separated by sodium dodecyl sulfate-polyacrylamide gel electrophoresis with 7.5% SuperSep Ace (FUJIFILM Wako Pure Chemical) and transferred to a polyvinylidene difluoride membrane (GE Healthcare, Menlo Park, CA) using a Trans-Blot SD Cell (Bio-Rad Laboratories). The membrane was blocked in 4% Block Ace (KAC Co., Kyoto, Japan), and then reacted with HA-Tag Rabbit mAb (clone C29F4, Cell Signaling Technology, Danvers, MA) in tris-buffered saline containing 0.05% Tween-20 and 1% bovine serum albumin. The membrane was then washed and reacted with anti-rabbit IgG HRP-linked antibody (Cell Signaling Technology). Immunoreactive proteins on the membrane were detected using ImmunoStar Zeta (FUJIFILM Wako Pure Chemical) and ImageQuant LAS4010 (GE Healthcare).

### 2.8. Cytotoxicity Assay

VEGFR2^−^ target cells (EL4 and L1.2 cells) were labeled with Tag-It Violet Proliferation Cell Tracking Dye (Biolegend), and VEGFR2^+^ target cells (mVEGFR2^+^ EL4 cells and hVEGFR2^+^ L1.2 cells) were labeled with Cell Proliferation Dye eFluor 670 (Thermo Fisher Scientific). Mouse CAR-T cells four days after Rv transduction or human CAR-T cells 24 h after electroporation, VEGFR2^−^ target cells, and VEGFR2^+^ target cells were co-cultured at the indicated effector cells-to-VEGFR2^+^ target cells (E/T) ratios. After 18 h, CountBright Absolute Cell Counting Beads (Thermo Fisher Scientific) and 7-AAD Viability Staining Solution (Biolegend) were added to the reaction wells, and the number of each target cell was analyzed using flow cytometry until 1000 beads were detected. Cytotoxicity was calculated using the following formula: percentage of antigen-specific lysis = [(VEGFR2^+^ target cells/VEGFR2^−^ target cells ratio in non-effector cell’s well) − (VEGFR2^+^ target cells/VEGFR2^−^ target cells ratio in effector cell’s well)]/[VEGFR2^+^ target cells/VEGFR2^−^ target cells ratio in non-effector cell’s well] × 100.

### 2.9. BrdU Proliferation Assay and Cytokine Enzyme-Linked Immunosorbent Assay (ELISA)

CAR-T cells four days after Rv transduction were cultured for 24 h on a plate coated with mVEGFR2-Fc (2–2000 ng/mL). Proliferation activity was measured by ELISA using BrdU (Sigma-Aldrich). The production of interferon-γ, tumor necrosis factor-α, and interleukin-2 in the supernatants was measured using the OptiEIA™ ELISA Set (BD Bioscience).

### 2.10. Statistical Analysis

All experimental data were represented as the mean ± *SD*. Statistical significance was evaluated using GraphPad Prism Software (GraphPad Software, San Diego, CA).

## 3. Results

### 3.1. Expression of Various Hinge Domain (HD)-Modified and Hinge/Transmembrane Domain (HD/TMD)-Modified CARs in Mouse T Cells

Since CAR HD/TMD are assumed to regulate CAR expression level, we hypothesized that differences in HD/TMD would affect the amount of CAR signal input to T cells. In this study, we aimed to reveal the role of HD/TMD in CAR activity and CAR-T cell function using a first-generation CAR that inputs only a CD3ζ signal, rather than a second-generation CAR that inputs the CD3ζ signal and the co-stimulatory signal that modulates it. We constructed a basic CAR (mV/3z/3z/3z), in which CD3ζ-derived HD/TMD/STD was linked to anti-mVEGFR2 scFv, and HD-modified and HD/TMD-modified CARs that replaced HD and HD/TMD of mV/3z/3z/3z with the corresponding regions of CD4, CD8α, and CD28 that are constitutively expressed on T cells (Figure 1A). Our previous work established a stable and highly efficient CAR gene T cell transduction protocol for analyzing CAR structure–activity relationships [15]. In the CAR-T cells generated in this study, CAR mRNA levels of all CAR variants were comparable to those of mV/3z/3z/3z, and were constant for at least six days after Rv transduction (Figure 2A). Therefore, HD or HD/TMD modification did not affect transcription from the CAR gene. In contrast, these CAR expression levels on T cells differed substantially (Figure 2B). Whereas replacement of mV/3z/3z/3z with CD4-HD did not change its surface expression level, replacement with CD8α-HD or CD28-HD caused CAR expression level to increase significantly immediately after Rv transduction (day 0), but then returned to mV/3z/3z/3z levels two days later. CAR expression level was enhanced much more in HD/TMD-modified than in HD-modified CARs, and decreased in the following order: mV/8a/8a/3z > mV/28/28/3z > mV/4/4/3z. Unlike HD-modified CARs, HD/TMD-modified CARs only gradually disappeared from the T cell membrane, with mV/8a/8a/3z and mV/28/28/3z exhibiting significantly higher surface expression levels than mV/3z/3z/3z, even six days after Rv transduction. Furthermore, the mVEGFR2-binding of these CARs correlated with CAR expression level, confirming that HD/TMD changes do not affect the mVEGFR2-recognition properties of the ARD (Figure 2C).

Next, western blot analysis was performed to compare the expression modality of CAR variants in T cells (Figure 2D). Instead of being expressed as a monomer, the basic structure existed mostly as a complex exceeding 130 kDa. The three HD-modified CARs showed almost no band of the predicted monomer size, and mV/8a/3z/3z appeared in two types of complexes with distinctly different molecular sizes. In comparison, the two HD/TMD-modified CARs (mV/4/4/3z and mV/8a/8a/3z) revealed increased monomer abundance and the intermingling of the monomer with the complex. Moreover, mV/28/28/3z showed the same complex formation as mV/28/3z/3z. In the reduced sample, the bands corresponding to the complex disappeared and, instead, a band of the predicted monomer size (46.4–52.5 kDa) was detected for each CAR variant. Therefore, bands exceeding 130 kDa detected under non-reducing conditions likely corresponded to CAR multimers or complexes of CAR with other membrane proteins linked together by cysteine-mediated disulfide bonds within HD/TMD (Figure 1C and [16,17,18,19]). Furthermore, mV/8a/3z/3z, mV/28/3z/3z, mV/8a/8a/3z, and mV/28/28/3z under reducing conditions showed clear bands of larger molecular size than the monomers, suggesting the presence of CARs that had undergone glycosylation within CD8α-HD [17] or CD28-HD [19] (Figure 1C).

Taken together, these results indicate that HD/TMD-modification of CAR does not significantly affect ARD affinity. Both the HD and TMD structure affect the CAR expression efficiency to the cell membrane, and the expression topology of CAR on T cells through glycosylation or formation of a complex via disulfide bonds. In addition, the stability of CAR expression on the membrane is mainly regulated by TMD.

### 3.2. Function of Mouse T Cells Expressing Various HD-Modified and HD/TMD-Modified CARs

We compared the antigen-specific cytotoxic activity of various structurally modified CAR-T cells cultured for four days after Rv transduction (Figure 3A). Among the HD-modified CAR-T cells, CAR [mV/28/3z/3z]-T cells showed higher cytotoxic activity than basic CAR [mV/3z/3z/3z]-T cells. Notably, all three HD/TMD-modified CAR-T cells showed enhanced cytotoxic activity in the following order: mV/28/28/3z > mV/8a/8a/3z > mV/4/4/3z. Although a positive correlation was observed between CAR expression level and cytotoxic activity of CAR-T cells, there was a significant difference in cytotoxic activity between mV/28/3z/3z and mV/8a/3z/3z or mV/28/28/3z and mV/8a/8a/3z, even though they showed similar surface expression level. This result suggests that CARs with CD28-HD have higher CD3ζ signal input efficiency associated with target antigen binding than CARs with CD8α-HD, despite them exhibiting the same surface expression efficiency and sustainability.

Next, we compared antigen density responsiveness in antigen-specific cell proliferation activity using HD/TMD-modified CAR-T cells, in which CAR expression persisted after Rv transduction (Figure 3B). CAR [mV/8a/8a/3z]-T cells and CAR [mV/28/28/3z]-T cells that showed high CAR expression proliferated in the presence of 2 ng/mL mVEGFR2-Fc, and reached a plateau at 20 ng/mL. Proliferation of CAR [mV/3z/3z/3z]-T cells and CAR [mV/4/4/3z]-T cells was similarly promoted with 2 ng/mL mVEGFR2-Fc, but maximum proliferative activity was achieved with 200 ng/mL mVEGFR2-Fc. A high positive correlation was observed between CAR expression level and antigen-specific proliferation of each CAR-T-cell type, suggesting that HD/TMD modification affected the proliferative activity of CAR-T cells by increasing or decreasing antigen stimulation intensity according to the CAR expression level.

Next, we examined dose dependence of antigen stimulation on the cytokine production ability of HD/TMD-modified CAR-T cells (Figure 3C). Unlike with proliferative activity, the antigen stimulation threshold required for cytokine production differed among HD/TMD-modified CAR-T cells. For interferon-γ and interleukin-2, the thresholds of the amount of mVEGFR2-Fc required for their secretion were lower for CAR [mV/8a/8a/3z]-T cells and CAR [mV/28/28/3z]-T cells than for CAR [mV/3z/3z/3z]-T cells and CAR [mV/4/4/3z]-T cells. For tumor necrosis factor-α, the threshold was lower for CAR [mV/28/28/3z]-T cells than for the other three CAR-T cell types. Moreover, although a positive correlation similar to that previously described for cytotoxic activity was observed between CAR expression level and various cytokine production levels of each CAR-T cell type, CAR [mV/28/28/3z]-T cells showed significantly higher cytokine-producing ability than CAR [mV/8a/8a/3z]-T cells despite an equivalent CAR expression level.

These results showed that the amount of CAR signal input associated with CAR antigen binding was highly dependent on the CAR surface expression level (expression stability on cell surface) due to the contribution of TMD. Moreover, the HD structure may directly affect the CAR signaling threshold (the efficiency of CD3ζ-STD signaling).

### 3.3. Expression and Function of HD/TMD-Modified CARs in Human T Cells

Given that the AA sequences of mouse and human immune molecules do not match completely, it is important to determine whether HD/TMD characteristics of mouse CAR can be extrapolated to human CAR. CD8α-HD/TMD shares 62.8% AA identity between mice and humans, whereas CD28-HD/TMD shares 66.8% AA identity (Figure 1C). We constructed two human second-generation CARs (hV/h8a/h8a/h28-h3z and hV/h28/h28/h28-h3z) with hCD8α-HD/TMD or hCD28-HD/TMD for clinical use (Figure 1B), and examined whether the structure–activity relationship of human CAR was similar to that of mouse CAR. Two kinds of human CAR-T cells were prepared by CAR mRNA electroporation [12]. Introduction efficiency and disappearance of CAR mRNA were equivalent between the two types (Figure 4A). Surface expression of CAR translated from mRNA was slightly, albeit not significantly, higher in hV/h8a/h8a/h28-h3z than in hV/h28/h28/h28-h3z (Figure 4B). As CAR mRNA disappeared, both CARs vanished from the T cell membrane with a half-life of approximately 12 h. These results confirm that each CAR with hCD8α- or hCD28-HD/TMD shows equivalent efficiency and sustainability of surface expression even in human T cells, despite a different production method from mouse CAR-T cells. Finally, the antigen-specific cytotoxic activity of human CAR-T cells was evaluated 24 h after electroporation. CAR [hV/h28/h28/h28-h3z]-T cells showed significantly higher cytotoxic activity than CAR [hV/h8a/h8a/h28-h3z]-T cells, and its cytotoxicity per CAR expression level was about 2.5 times higher (Figure 4C). Therefore, in both human and mouse CAR-T cells, HD/TMD modification affects the CAR surface expression level as well as CAR activity.

## 4. Discussion

Although CAR-T cell therapy has proven to be a breakthrough therapy with potent and sustained antitumor activity, the occurrence of serious side effects and the low efficacy in solid tumors have gradually become apparent as a challenge. Therefore, many researchers are investigating the possibility of improving the function and persistence of CAR-T cells by modifying the CAR structure, establishing a CAR-T cell manufacture protocol, and combining it with existing anti-cancer medicine [20,21,22]. CAR is an artificial receptor with an ARD and an intracellular T cell activation motif, and there are no clear rules or guidance for its structural design. Although the antigenicity of CAR should be considered carefully, various membrane proteins are used as templates to improve the CAR structure. However, most CARs developed to date have components in a defined pattern, and designing CAR constructs for individual diseases and target antigens has not been studied satisfactorily [23,24]. Research on the CAR structure has focused mainly on the ARD, which is responsible for antigen specificity [25,26,27], and STD, which translates activation signals [27,28,29,30]. In contrast, the HD/TMD of most CARs empirically selected CD8α- or CD28-derived sequences as structures that can be efficiently expressed on T cells, and their role in CAR activity and CAR-T cell functions are poorly understood. Here, we demonstrated that differences in the HD/TMD structure alter surface expression level and activity of CAR, and found that the HD/TMD-modification could adjust the functional strength of CAR-T cells without impairing the antigen binding properties of ARD and the signaling properties of STD to modify T cell functions. 

Our analysis of CAR expression suggested that HD affected the transport efficiency of CAR proteins to the cell membrane (Figure 5A) and that TMD regulated the membrane expression stability of CAR (Figure 5B). The transport rate of a nascent protein depends on the efficiency with which the protein adopts a thermodynamically stable structure in the secretory pathway [31]. Therefore, the membrane transport rate of each CAR is speculated to depend on the folding efficiency of the CAR extracellular domain, that is, the efficiency of immunoglobulin domain formation in scFv. Recent studies have reported different expression levels of CAR depending on scFv structure, even in the same HD/TMD/STD structure [14,32], and we have shown that the CAR surface expression efficiency is affected by differences in the structural stability of scFv [14]. In the present study, a side-by-side comparison of some CARs, each with a common anti-VEGFR2 scFv with high structural stability, suggests that the different membrane transport efficiencies of CAR proteins had a significant effect on their HD structure. In particular, the number of amino acids in HD (CD8α; 65 AA > CD28; 36 AA > CD4; 23 AA > CD3ζ; 9 AA) positively correlated with the CAR expression level, strongly suggesting that the folding efficiency of CAR scFv depends on the HD length. Furthermore, as complex formation via HD/TMD and glycosylation in HD are expected to contribute to the structural stabilization of the CAR extracellular domain, these post-translational modifications would also be the factors influencing the membrane transport rate of CARs.

In contrast, differences in the TMD structure altered CAR surface expression levels and stability, but did not affect the CAR mRNA level or total amount of CAR protein, suggesting that TMD regulates the CAR expression stability on cell surface as well as its cellular localization. The previous study reported that endogenous CD3ζ disappear from the T cell membrane in a few hours [13], consistent with the low expression stability of CARs with CD3ζ-TMD. In addition, recent reports indicate that endocytosis causes CD3ζ to disappear from the cell membrane, translocate to recycling endosomes, and be transported back to the cell membrane in response to CD3 signaling via the T cell receptor [33]. Although research on the intracellular transport of proteins has been actively conducted in recent years, the factors that affect the transport and localization of the membrane protein by its TMD structure have not been elucidated. Future designs of the CAR TMD may take into account the intracellular dynamics of CAR such as intracellular translocation associated with antigen stimulation and re-expression through a recycling pathway.

Functional analysis of various structurally modified CAR-T cells has revealed that the differences in HD structure changes the CAR activity. The antigen-recognized CAR transduces the T cell activated signal by exposing the immunoreceptor tyrosine-based activation motifs of CD3ζ-STD, which are otherwise protected on the cell membrane [34]. Given that the exposure efficiency of CD3ζ-STD is supposed to depend on the degree of CAR structural change, the efficiency of CD3ζ-STD signaling was assumed to alter by HD structural properties or CAR oligomerization. The previous studies of CAR with immunoglobulin domain-derived HD has shown that the CAR-T cell functions is enhanced by extending the HD length [25,35,36]. However, the present analysis using CAR with immune molecule-derived HD showed that HD length (CD8α > CD28 > CD4 > CD3ζ) did not correlate with the functional strength of CAR-T cells (CD28 > CD3ζ, CD4, CD8α). This is because the HD used in the present study does not form a tertiary structure akin to that of the immunoglobulin domain. In many immune molecule-derived HDs, the random coil sequences from the TMD to characteristic extracellular domain of its original molecule is adopted. However, only CD28-HD has a β-sheet forming motif that is part of the immunoglobulin domain (Figure 1C). We speculate that CAR with CD28-HD is less flexible in the extracellular region than CARs, whose HDs are derived from other molecules, and CAR with CD28-HD is likely to change its structure when the ARD binds to an antigen.

CAR oligomerization will also be a factor in changing the efficiency and quantity of CD3ζ-STD signaling. CD3ζ, CD8α, and CD28 used for the CAR HD/TMD in this study are originally present on T cells as dimers, and only CD4 is expressed as a monomer. Cysteine residues responsible for oligomerization are present in CD3ζ-TMD [13], CD8α-HD [17], and CD28-HD [18] (Figure 1C), which presumably contribute to CAR oligomerization. Furthermore, previous studies using human T-cell leukemia cell lines (Jurkat cells) have reported that CARs with CD3ζ-TMD form heterodimers with endogenous CD3ζ and the presence of this complex enhances the functionality of CAR-expressing Jurkat cells [37]. Here, too, some CARs (mV/3z/3z/3z and mV/4/3z/3z) with CD3ζ-TMD had a size of approximately 60 kDa and are assumed to be heterodimers of CAR (46–48 kDa) and endogenous CD3ζ (15 kDa). Therefore, it is highly likely that CAR and endogenous molecules form a complex in primary T cells. On the other hand, cysteine residues in HD/TMD are also anticipated to form unexpected complexes with membrane proteins different from its original molecule or same CAR molecule, as detected in multiple bands in western blot analysis. Depending on the type and size of such membrane proteins, an unexpected signal may be transduced together with the CAR signal, or the antigen-binding properties of CAR may be lost due to steric hindrance. Therefore, identifying the components of the CAR complex using mass spectrometry and the functional analysis of CAR with mutants of amino acids involved in post-translational modification remains an important future task. 

Investigating the available information on HD length and flexibility, and the CAR complex formation proposed in this study is expected to provide HD/TMD structural information for CAR design and facilitate the discovery of promising candidate molecules or sequences beyond the existing frameworks as CAR HD/TMD. Furthermore, the analysis of the combination of HD/TMD and STD is also an important consideration to apply the structural information of HD/TMD to the second- and third-generation CARs for clinical use because the previous study has shown that TMD derived from the same molecules as STD proximal to the plasma membrane is more efficient for CAR signal input [29]. Considering that the tumor necrosis factor receptor superfamily (TNFRSF) such as 4-1BB and OX40 are signaled by trimerization [38], a design of HD/TMD that induces CAR trimerization may improve the efficiency of TNFRSF-derived STD signaling as many second-generation CARs with 4-1BB-STD incorporate CD8α-HD/TMD instead of CD28-HD/TMD [6,8]. Thus, identifying the elements of HD and TMD that affect the mechanism of CAR expression and signal input will greatly contribute to establishing of the CAR design methodology for generating CAR-T cells with potent antitumor activity without severe side effects.

The in vivo analysis focusing on the antitumor effects and persistence of CAR-T cells is essential for creating a clinically effective CAR-T cell medicine because the in vitro cytotoxic activity of CAR-T cells does not always correlate with their in vivo efficacy [39]. Unfortunately, none of the HD/TMD-modified CAR variants used in this study showed significant antitumor activity as tumor vessel-injuring CAR-T cells because those CARs were first generation molecules, and therefore we were unable to evaluate their function in vivo. We are proceeding with the analysis of second generation CARs with different HD/TMD, and are establishing a hematological malignancy model and a solid tumor model to analyze the relationship between CAR structure and CAR-T cell in vivo function. Thus, further research on the relationship between CAR structure and CAR-T cell function by making full use of the mouse CAR-T cells will lead to the creation of promising novel CAR constructs based on scientific evidence, and should accelerate the development of highly effective and safe human CAR-T cell medicine.

## 5. Conclusions

CAR HD and TMD affected the amount of CAR signaling in T cells via the regulation of the CAR avidity and activity. HD played a role in the control of the modality of CAR expression and membrane transport efficiency of CAR, and the definition of the CAR signaling threshold (Figure 5A). TMD played a role in the regulation of the amount of CAR signaling into T cells via the control of the CAR surface expression level (Figure 5B). By performing CAR structure–activity relationship analysis focusing on HD/TMD, it will be possible to adjust the functional strength and the response to antigen density of CAR-T cells, while retaining the antigen specificity of the ARD and the signal characteristics of the STD (Figure 5C).

## Figures and Tables

**Figure 1 cells-09-01182-f001:**
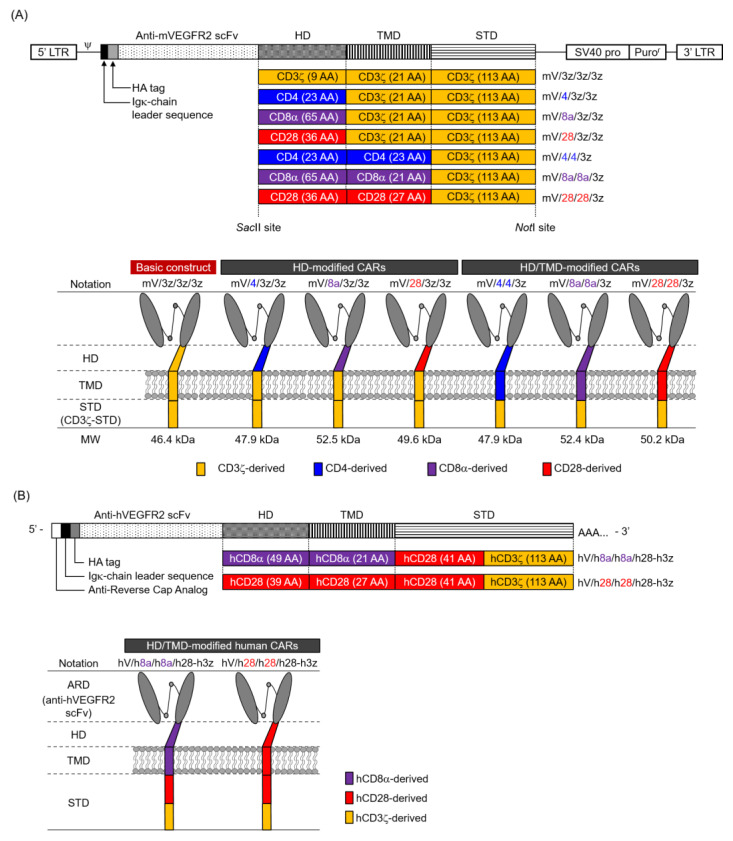
Illustration of vascular endothelial growth factor receptor 2 (VEGFR2)-specific chimeric antigen receptors (CARs) with hinge domain (HD) or hinge/transmembrane domains (HD/TMD) modification. (**A**) Top panel, retroviral vector constructs containing the murine VEGFR2-specific first-generation CAR genes for mouse T cells; lower panel, representation of HD-modified or HD/TMD-modified CAR proteins. The figure shows the number of amino acids (AA) used to generate HD/TMD/signal-transduction-domain (STD) and the molecular weight (MW) of each CAR. (**B**) Top panel, human VEGFR2-specific second-generation CAR mRNA constructs for human T cells; lower panel, representation of HD/TMD-modified CAR proteins. (**C**) AA sequences of CD3ζ, CD4, CD8α, and CD28 used for constructing the HD/TMD in this study. Cysteines that form interchain disulfide bonds are shown in bold and glycosylation sites are shown in bold italic. For CD8α-HD/TMD and CD28-HD/TMD, sequence alignment between mouse and human is shown. An asterisk denotes an identical AA, whereas alignment gaps are indicated by a hyphen. In CD28-HD, the area that forms the β-sheet structure is underlined.

**Figure 2 cells-09-01182-f002:**
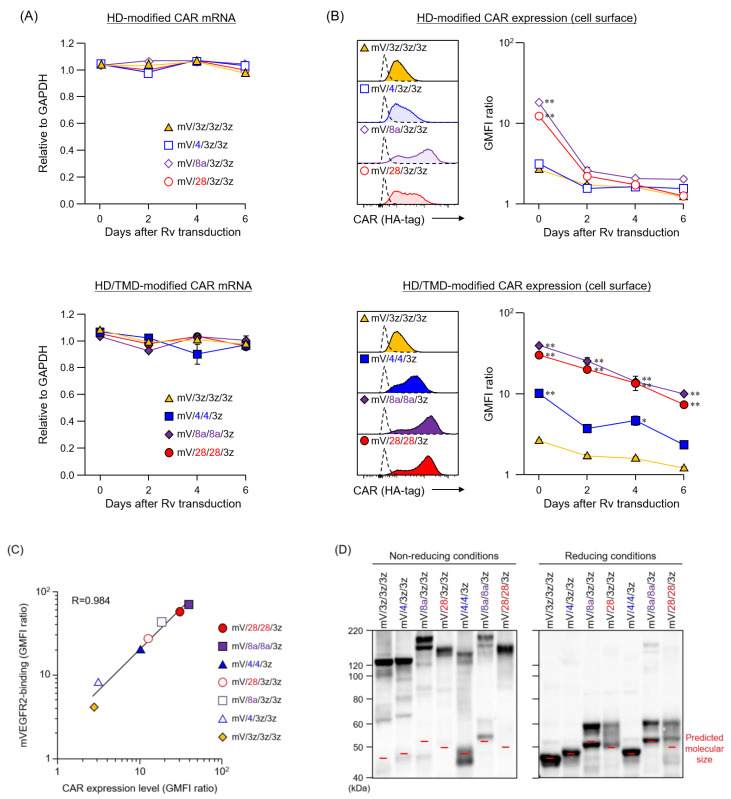
Expression of hinge domain (HD)-modified and hinge/transmembrane domain (HD/TMD)-modified chimeric antigen receptors (CARs) in mouse T cells. (**A**) CAR mRNAs were analyzed by reverse transcription and quantitative polymerase chain reaction, and their transcription levels were calculated relative to GAPDH mRNA as an endogenous control. (**B**) CAR expression on T cells were analyzed by flow cytometry using anti-HA-tag mAb (solid color histograms) or isotype control antibody (dashed white histograms). Each CAR expression level was calculated from the ratio of geometric mean fluorescence intensity (GMFI) when stained with the anti-HA-tag mAb to GMFI when stained with the isotype control antibody. (**C**) The mouse VEGFR2-binding of CAR-T cell on day 0 was analyzed by flow cytometry using an mVEGFR2-Fc chimera, and the relationship between the CAR expression level and mVEGFR2-binding of CAR-T cells was evaluated. (**D**) CAR proteins in the whole lysate were analyzed by sodium dodecyl sulfate-polyacrylamide gel electrophoresis and western blot analysis using anti-HA-tag mAb on day 0. The data were obtained from three independent tests. Statistical analysis was performed using Dunnett’s test against mV/3z/3z/3z, * *p* < 0.05; ** *p* < 0.01.

**Figure 3 cells-09-01182-f003:**
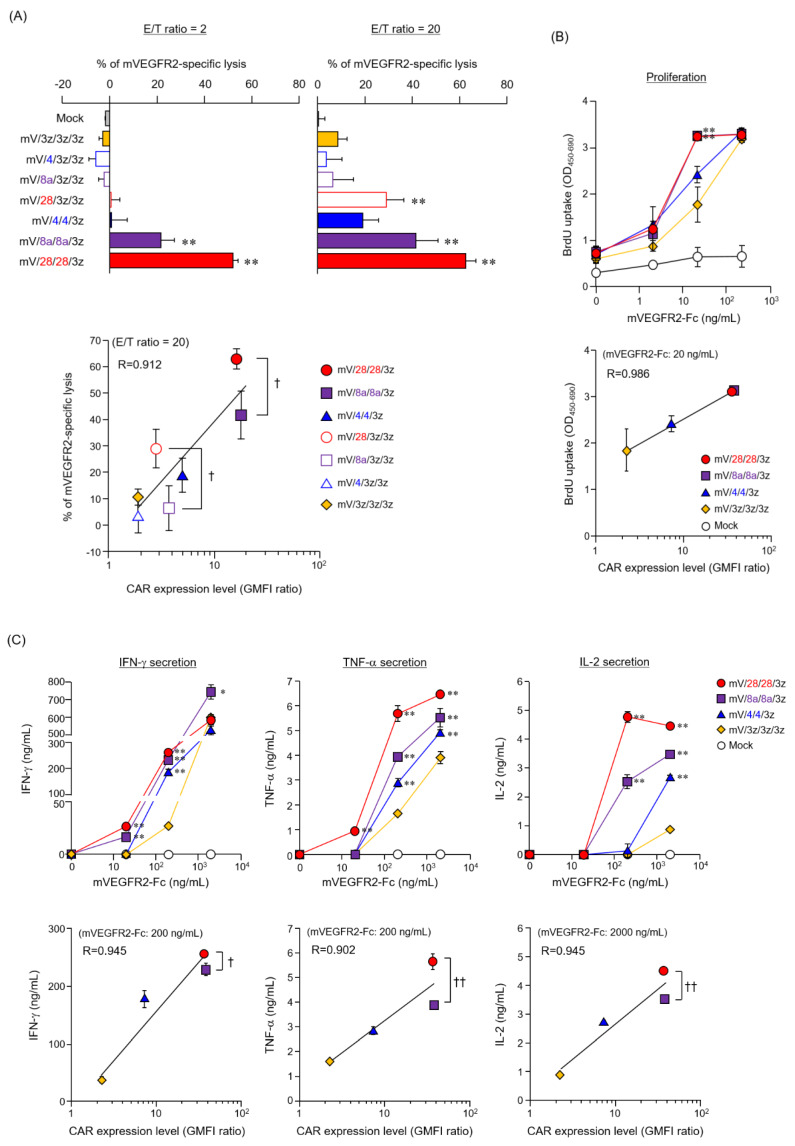
Functional characteristics of mouse T cells expressing CARs with different hinge domain (HD) and transmembrane domain (TMD). (**A**) Top panel, cytotoxic activity of HD-modified and HD/TMD-modified CAR-T cells four days after Rv transduction against mVEGFR2^+^ EL4 cells; lower panel, relationship between cytotoxicity at E/T ratio = 20 and CAR expression level. (**B**) Top panel, proliferation activity of HD/TMD-modified CAR-T cells following mVEGFR2-stimulation; lower panel, relationship between proliferation activity upon stimulation with mVEGFR2-Fc 20 ng/mL and CAR expression level. (**C**) Top panel, cytokine-producing ability of the above cells: interferon-γ, tumor necrosis factor-α, and interleukin-2; lower panel, relationship between cytokine-producing ability upon stimulation with mVEGFR2-Fc 200 or 2000 ng/mL and CAR expression level. The data are representative of at least two independent tests. Statistical analysis was performed using Dunnett’s test against mV/3z/3z/3z; * *p* < 0.05, ** *p* < 0.01; or Student’s *t*-test; ^†^
*p* < 0.05, ^††^
*p* < 0.01.

**Figure 4 cells-09-01182-f004:**
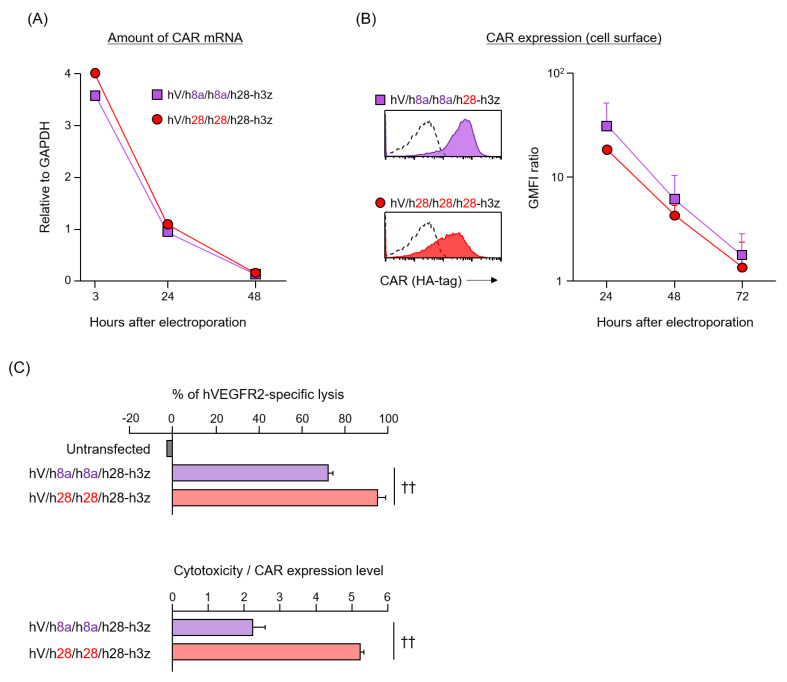
Expression and function of hinge/transmembrane domains (HD/TMD)-modified CARs in human T cells introduced CAR mRNA by electroporation. (**A**) CAR mRNAs were analyzed by reverse transcription and quantitative polymerase chain reaction, and their amount were calculated relative to GAPDH mRNA. (**B**) CAR expression on T cells were analyzed by flow cytometry using anti-HA-tag mAbs (solid color histograms) or isotype control antibody (dashed white histograms). Each CAR expression level was calculated from the ratio of geometric mean fluorescence intensity (GMFI) when stained with the anti-HA-tag mAb to GMFI when stained with the isotype antibody. (**C**) Top panel, cytotoxic activity of human CAR-T cells cultured for 24 h after electroporation against hVEGFR2^+^ L1.2 cells at E/T ratio = 10; lower panel, cytotoxicity per CAR expression level. The data were obtained from three independent tests. Statistical analysis was performed using Student’s *t*-test; ^††^
*p* < 0.01.

**Figure 5 cells-09-01182-f005:**
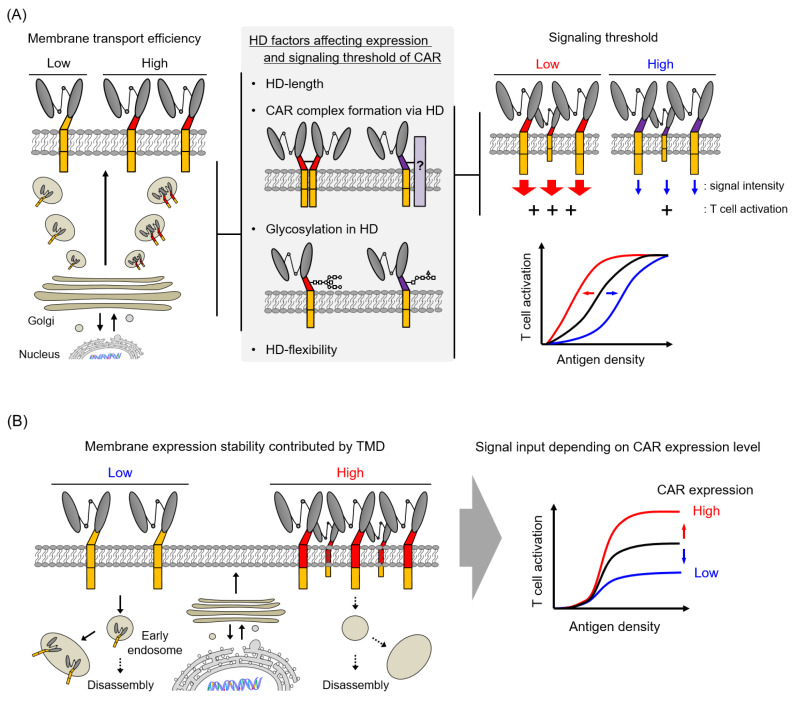
Role of hinge domain (HD) and transmembrane domain (TMD) in CAR. (**A**) Role of HD in the expression modality and membrane transport efficiency as well as CAR signaling threshold. (**B**) Role of TMD in CAR intracellular dynamics and surface expression stability. (**C**) Strategy to adjust CAR-T cell function by modifying CAR HD/TMD.

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
