# Peer review of "Hinge and Transmembrane Domains of Chimeric Antigen Receptor Regulate Receptor Expression and Signaling Threshold"

_cells, 2020, doi:10.3390/cells9051182_

Round 1
Reviewer 1 Report
Major limitations and issues:
- The title of the paper “Hinge and transmembrane domains of chimeric antigen receptor regulate receptor expression and activation threshold” suggest that the conclusions from the paper can be applied to all CARs. However, it has long been known that the antigen recognition domain also influences the CAR behavior und only the VEGFR2 scFv is used in this study. Thus, the conclusions from this study could only be applied to CARs containing the exact VEGFR2 scFv and therefore, the title of the paper needs to be changed.
- The general conclusion from this paper is that extracellular spacer and transmembrane domains affect the expression pattern of CARs. The connection between spacer and transmembrane domain and CAR expression has already been established (reviewed in Stoiber et al.; Cells; 2019; Guedan et al.; Molecular Therapy Methods & Clinical Development). Thus, this concept is not novel.
- In their paper, the authors focus mainly on murine CAR constructs. Only in the very last experiment, human CARs with different modifications are evaluated, but the data, that is provided is very limited. In order to make conclusion for CARs to be used in humans, the majority of experiments should be performed with human CARs
- The authors are drawing conclusions from several experiments that are not conclusive and/or that are speculative and not supported by the provided data:
- Figure 2D, page 8, line 238: “Furthermore the HD changes the expression topology of CAR on T cells through glycosylation or formation of a complex via disulfide bonds”: it requires further experiments, e.g. by introducing mutations into the sequence to support this theory and the contribution of the TMD was not taken into consideration
- Figure 3A, page 8, line 247: “…, there was a significant difference between in cytotoxic activity between mV/28/3z/3z and mV/8a/3z/3z or mV/28/28/3z and mV/8a/8a/3z, even though they showed similar surface expression levels.” This conclusion cannot be drawn for the figure since the significance was tested against mV/3z/3z/3z and it seems very unlikely that there is a statistical significance between mV/28/3z/3z and mV/8a/3z/3z.
- Figure 3A, page 8, line 249: “This result suggests that CARs with CD28-HD have higher CD3z signa input efficiency associated with target antigen binding than CARs with CD8a-HD…”: Again, no data is provided to support this hypothesis. It is also possible that the different cytotoxicity capacity in the short-term assay is caused by differences in the phenotype of the T cells, but this was not considered by the authors.
- Figure 3C, page 10, lines 274-286: The conclusion of this section is very imprecise. The conclusion is “…, the HD structure may directly affect the CAR signal input efficiency.” First, the term signal input efficiency needs a precise explanation and it remains absolutely unclear what the direct effect could be. This needs to be explained in more detail.
- In figure 5A, the authors are stating that the activation threshold depends on the structural characteristics of the HD. However, this is not shown in the provided data. In figure 3C, the authors show the correlations between antigen density and cytokine secretion for HD and TMD modified CAR constructs only.
- In the graph of figure 5B, the authors are claiming that the intenisity of T cell activation at a certain antigen density depends on the level of car expression level. Again, they do not clearly show data for this assumption in the paper.
- The article lacks explanation of terms that are very important to be clarified: e.g. activation threshold, signal input efficiency. The authors are using those terms, but the meaning doesn’t become clear for the reader.
- The authors are using short-term cytotoxicity assays in order to evaluate the function of CAR T cells. However, CAR T cell function does not only depend on the cytotoxic capacity but rather on the combination of proliferation, persistence and cytotoxicity. Therefore, long-term killing assays that evaluate cytotoxicity and proliferation would be more appropriate.
- The discussion part also needs to be re-structured. In general, this discussion is not focused on the results presented in the paper. It is rather a listing of several assumptions, that are not adequately cited in some cases or the cited publication does not support the assumption.
Minor issues
- Labeling in the figures should be modified in a way that every condition has the same symbol in all figures. This would help
- Page 7, line 222: authors are referring to figure 2C but the western blot analysis is shown in figure 2D
- Line 318: This is not precise because there have been publications that highlight the importance of of spacer and transmembrane domains.
- Line 337: “positive correlation between CAR expression and the number of amino acids in HD”: this was not the case for the human CAR constructs because no difference was detectable in the expression level between CD8a- and CD28 derived CARs (figure 4B)
- Line 344: the conclusion what the exact role of the TMD is could only be drawn if constructs were included that differed with respect to their TMD only. Otherwise it is always possible that the HD also contributes to the effect.
Author Response
Reply to the Review Report (Reviewer 1):
We appreciate you taking the time to offer us your comments and insights related to the paper. We found your feedback very constructive and tried to be responsive to your concerns. Our responses are given in a point-by-point manner below. Changes in the manuscript are underlined and highlighted in yellow.
Q1;
The title of the paper “Hinge and transmembrane domains of chimeric antigen receptor regulate receptor expression and activation threshold” suggest that the conclusions from the paper can be applied to all CARs. However, it has long been known that the antigen recognition domain also influences the CAR behavior und only the VEGFR2 scFv is used in this study. Thus, the conclusions from this study could only be applied to CARs containing the exact VEGFR2 scFv and therefore, the title of the paper needs to be changed.
A1;
To reinforce the conclusions of this paper, we also understand that further analysis using CARs with scFv for antigens other than VEGFR2 is needed. In your comment, does “the antigen recognition domain also influences the CAR behavior” imply that the surface expression efficiency of CAR changes depending on the type of scFv? If correct, it is strongly assumed that the change in expression efficiency between CARs incorporating different scFv is due to differences in the structural stability of scFv. We have already published the results of our research in this regard in Biochemical and Biophysical Research Communications (in press, DOI: 10.1016/j.bbrc.2020.03.071). Based on these findings, the following text and references were inserted into the Discussion.
Page13, lines 344-351:
“We have already shown that the surface expression efficiency of CAR varies with the structural stability of the scFv used for ARD of CAR [14]. In this study, a side-by-side comparison of some CARs, each with a common anti-VEGFR2 scFv with high structural stability, suggests that the different membrane transport efficiencies of the CARs had a significant effect on their HD structure. In particular, the number of amino acids in HD (CD8a; 65 AA > CD28; 36 AA > CD4; 23 AA > CD3z; 9 AA) positively correlated with the CAR expression level, strongly suggesting that the folding efficiency of CAR scFv depends on the HD length.”
In this study, we used two anti-VEGFR2 scFvs with high structural stability in the ARD of CAR, and found that the CAR expression and activity depends on their HD/TMD structure. We believe that our conclusion that the structure of HD and TMD affects the expression and signaling mechanism of "CAR proteins" applies to all CARs with structurally stable scFv. Therefore, we have determined that the title of this paper is reasonable and does not need to be changed.
Q2;
The general conclusion from this paper is that extracellular spacer and transmembrane domains affect the expression pattern of CARs. The connection between spacer and transmembrane domain and CAR expression has already been established (reviewed in Stoiber et al.; Cells; 2019; Guedan et al.; Molecular Therapy Methods & Clinical Development). Thus, this concept is not novel.
A2;
We do not believe that “the connection between the spacer and transmembrane domain and CAR expression” has been clarified in detail in previous studies. Stoiber et al. and Guedan et al. have summarized the research results on the effects of four regions of CAR on CAR-T cell function, but did not report the establishment of a design method for HD/TMD to control CAR expression. Indeed, Stoiber et al. write in the text that " Although CD8α and CD28 spacers have been extensively utilized in numerous clinical anti-CD19-CAR trials demonstrating potent anti-tumor efficacy, the impact of these hinges on CAR T cell performance has not been thoroughly investigated as of yet, thus their use remains largely empirical. " (3. Spacer domain, fourth paragraph). In our paper, the structural elements of HD/TMD that regulate the CAR expression mechanism have not been clarified, but we found for the first time that HD length and TMD sequence will affect the membrane expression efficiency, membrane stability, and intracellular localization of CAR by comparing CARs with different HD/TMD in a side-by-side comparison. These results will provide useful basic information for the establishment of an approach to regulate CAR-T cell function by structural modification of CAR.
Q3;
In their paper, the authors focus mainly on murine CAR constructs. Only in the very last experiment, human CARs with different modifications are evaluated, but the data that is provided is very limited. In order to make a conclusion for CARs to be used in humans, the majority of experiments should be performed with human CARs
A3;
We understand that analysis in human CAR-T cells is important and that many CAR researchers want information about human CAR. On the contrary, knowing that the amino acid sequences of immune molecules used in HD/TMD are slightly different between human and mouse, we carried out the evaluation with mouse CAR-T cells because it is possible to produce consistent quality CAR-T cells in a short period of time by the highly efficient Rv-transduction protocol, regardless of the CAR structure. In other words, the use of mouse T cells enables high-throughput evaluation of a large number of CARs modified with ARD, HD, TMD, or STD. Another advantage is that mouse CAR-T cells can be evaluated in a tumor-bearing immunocompetent mouse model for their interaction with the host immune system, which is important for the development of CAR-T cells with high efficacy and safety. Unfortunately, none of the HD/TMD-modified CAR variants used in this study showed significant antitumor activity as tumor vessel-injuring CAR-T cells because those CARs were first-generation molecules, and therefore we were unable to evaluate their function in vivo. We are now establishing a blood cancer model and a solid cancer model to analyze the relationship between CAR structure and CAR-T cell function in vivo. For these reasons, we believe that the development of human CARs on the basis of the knowledge obtained in mouse CARs will lead to the rapid creation of truly promising human CAR-T cell medicine. These our intentions in this study have been added to the Discussion (page 14, lines 415-420).
Q4;
Figure 2D, page 8, line 238: “Furthermore the HD changes the expression topology of CAR on T cells through glycosylation or formation of a complex via disulfide bonds”: it requires further experiments, e.g. by introducing mutations into the sequence to support this theory and the contribution of the TMD was not taken into consideration
A4;
We are sorry that we have failed to take into account the contribution of TMD in our current text. Given the strong expectation that CAR forms a complex, at least via CD3z-TMD, we have revised the text below.
Page 8, lines 236-239:
“Both the HD and TMD structure affect the CAR expression efficiency to the cell membrane and the expression topology of CAR in T cells through glycosylation or formation of a complex via disulfide bonds. In addition, the stability of CAR expression on the membrane is mainly regulated by TMD.”
In addition, we agree that the impact of post-translational modifications in HD/TMD on the CAR behavior needs to be further investigated. It is entirely possible that glycosylation and complex formation could affect the surface expression stability and signal input mechanism of CAR. Therefore, we are now constructing CARs containing mutants of amino acids involved in post-translational modification and plan to analyze their expression and activity. We have added the following text to the Discussion.
Page 14, lines 397-399:
“Therefore, identifying the components of the CAR complex using mass spectrometry and the functional analysis of CAR with mutants of amino acids involved in post-translational modification remains an important future task.”
Q5;
Figure 3A, page 8, line 247: “…, there was a significant difference between in cytotoxic activity between mV/28/3z/3z and mV/8a/3z/3z or mV/28/28/3z and mV/8a/8a/3z, even though they showed similar surface expression levels.” This conclusion cannot be drawn for the figure since the significance was tested against mV/3z/3z/3z and it seems very unlikely that there is a statistical significance between mV/28/3z/3z and mV/8a/3z/3z.
A5;
Please take a look at the diagram below in Figure 3A. This graph shows the correlation between the intensity of CAR expression and cytotoxic activity in CAR-T cells, indicating that the activity of each CAR-T cell depends on the intensity of CAR expression. Not only that, but the cytotoxic activity was found to be significantly different between mV/28/3z/3z and mV/8a/3z/3z or mV/28/28/3z and mV/8a/8a/3z, despite the similar surface expression level. Of course, there is a statistically significant difference in the results of the cytotoxic activity of these CAR-T cells, which is shown by † (daggers, Student’s t-test; †p < 0.05) in the figure. Please reconfirm.
Q6;
Figure 3A, page 8, line 249: “This result suggests that CARs with CD28-HD have higher CD3z signal input efficiency associated with target antigen-binding than CARs with CD8a-HD…”: Again, no data is provided to support this hypothesis. It is also possible that the different cytotoxicity capacity in the short-term assay is caused by differences in the phenotype of the T cells, but this was not considered by the authors.
A6;
The cytotoxic activity of T-cells expressed CAR with CD28-HD was clearly higher than that of T-cells expressed CAR with CD8a-HD (Figure 3, please confirm.). In this study, all CARs are first-generation structures, and all CARs on T cells can bind to antigens (Figure 2C). Therefore, the strength of the CAR-T cell function is inferred to be the amount of CD3z-STD signaling. Although the differentiation status of each CAR-T cell was not clarified by surface marker analysis of T cells in this study, it is unlikely that the difference in the functional strength of CAR-T cells is due to the difference in T cell phenotype, because each CAR-T cell was cultured for only 4 days after Rv-transduction under T-cell activation conditions using anti-CD3/CD28 antibodies, and no abnormal mitotic proliferation or cell death due to the CAR tonic signaling was observed during the culture period. Therefore, we believe that the interpretation of this result is reasonable.
Q7;
Figure 3C, page 10, lines 274-286: The conclusion of this section is very imprecise. The conclusion is “…, the HD structure may directly affect the CAR signal input efficiency.” First, the term signal input efficiency needs a precise explanation and it remains absolutely unclear what the direct effect could be. This needs to be explained in more detail.
A7;
We apologize for not conveying the conclusions in Figure 3 in an easily understood manner. The "signal input efficiency" indicates the efficiency of CD3z-STD signaling (exposure of CD3z-STD to the cytoplasm) in the antigen-recognized CAR. We corrected the “signal input efficiency” to the “the efficiency of CD3z-STD signaling” or “CAR signaling threshold” to avoid any misunderstanding. In addition, we believe that the factors that directly affect the efficiency of CD3z-STD signaling are the formation of complexes via HD and the flexibility of HD, as described in the following text in Discussion.
Page 13, lines 366-399:
・CAR transduces the T cell activated signal by exposing the activation motifs of CD3z-STD.
・The exposure efficiency of CD3z-STD is supposed to depend on the degree of CAR structural change.
・Only CD28-HD has a b-sheet forming motif that is part of the immunoglobulin domain (Figure 1C). We speculate that CAR with CD28-HD is less flexible in the extracellular region than CARs, whose HDs are derived from other molecules, and CAR with CD28-HD is likely to change its structure when the ARD binds to an antigen.
・CD3z, CD8a, and CD28 are originally present on T cells as dimers. Cysteine residues responsible for oligomerization are present in CD3z-TMD [13], CD8a-HD [14], and CD28-HD [15] (Figure 1C)
・Previous study reported that CARs with CD3z-TMD form heterodimers with endogenous CD3z and the presence of this complex enhances the functionality of CAR-expressing Jurkat cells [32].
・identifying the components of the CAR complex using mass spectrometry and the functional analysis of CAR with mutants of amino acids involved in post-translational modification remains an important future task.
Q8;
In figure 5A, the authors are stating that the activation threshold depends on the structural characteristics of the HD. However, this is not shown in the provided data. In figure 3C, the authors show the correlations between antigen density and cytokine secretion for HD and TMD modified CAR constructs only.
A8;
In response to the seventh question, we modify the “activation threshold” to the “CAR signaling threshold”.
Again, the results of this study suggested that the HD structure affects the CAR signaling threshold. As shown in the bottom graph of Figure 3B and 3C, it is clear that the intensity of T cell activation depends on the CAR expression level. On the contrary, the intensity of T cell activation varies depending on the HD structure, as in the case of CAR with CD28-HD and CAR with CD8a-HD, even though their CAR expression levels are equivalent. These results were similar to those of the cytotoxicity assay, suggesting that the structure of HD is an important factor in influencing the CAR signaling threshold.
More detailed analysis is needed to identify the elements of the HD structure, for example; the HD length and flexibility or CAR-complex formation via HD, that affect the CAR signaling threshold.
Q9;
In the graph of figure 5B, the authors are claiming that the intensity of T cell activation at a certain antigen density depends on the level of car expression level. Again, they do not clearly show data for this assumption in the paper.
A9;
Please look at the bottom graph in Figure 3A-C. Our data shows that the cytotoxic activity, proliferative activity, and cytokine secretion in CAR-T cells all correlate positively with the intensity of CAR expression. Therefore, we concluded that “the intensity of T cell activation at a certain antigen density depends on the level of car expression level.”
Q10;
The article lacks an explanation of terms that are very important to be clarified: e.g. activation threshold, signal input efficiency. The authors are using those terms, but the meaning doesn’t become clear for the reader.
A10;
We apologize for not conveying our intentions properly. Again, “activation threshold” and “signal input efficiency” both refer to the efficiency of CD3z-STD signaling into T cell. We have corrected those two words to “the CAR signaling threshold” or “the efficiency of CD3z-STD to avoid any misunderstanding.
Q11;
The authors are using short-term cytotoxicity assays in order to evaluate the function of CAR T cells. However, CAR T cell function does not only depend on the cytotoxic capacity but rather on the combination of proliferation, persistence, and cytotoxicity. Therefore, long-term killing assays that evaluate cytotoxicity and proliferation would be more appropriate.
A11;
We fully understand that long-term survival of administered T cells in vivo is important for the efficacy of CAR-T cell therapy, and that long-term cytotoxicity studies and in vivo studies, in which the cytotoxic activity and proliferative capacity can be analyzed in a complex manner, are essential for the creation of highly effective CAR-T cell medicine. As noted in our reply to Q3, the analysis of the combination of proliferation, persistence, and cytotoxicity of CAR-T cells seems to be more appropriate for second-generation CAR-T cells than for first-generation CAR-T cells. Therefore, we are currently constructing CARs with different HD/TMD and STD, and are planning experiments including the CAR expression and CAR-T cells in vivo antitumor activity and persistence.
Q12;
The discussion part also needs to be restructured. In general, this discussion is not focused on the results presented in the paper. It is rather a listing of several assumptions, that are not adequately cited in some cases or the cited publication does not support the assumption.
A12;
In this paper, we discussed the mechanism of CAR expression and signal input based on the results obtained, and summarize the issues to be examined in the future. I think the content of the Discussion has been enriched by additions and revisions based on the points made by you. If there are any areas that are not yet appropriate, please elaborate.
Q13;
Labeling in the figures should be modified in a way that every condition has the same symbol in all figures. This would help
A13;
The symbols in each figure are assigned according to the HD/TMD structure of CAR. These symbols are unified in all figures.
Q14;
Page 7, line 222: authors are referring to figure 2C but the western blot analysis is shown in figure 2D
A14;
Thank you for pointing out the serious mistake. We have corrected the figure number on line 221 on page 7.
Q15;
Line 318: This is not precise because there have been publications that highlight the importance of spacer and transmembrane domains
A15;
We agree that the length of the spacer is important for ARD binding to the antigen. However, the role of HD (spacer) on CAR activity and CAR-T cell function, as shown in this paper, is poorly understood. Therefore, we believe that the text of line 318 (in the revised manuscript, line 329) is a reasonable representation.
Q16;
Line 337: “positive correlation between CAR expression and the number of amino acids in HD”: this was not the case for the human CAR constructs because no difference was detectable in the expression level between CD8a- and CD28 derived CARs (figure 4B)
A16;
The surface expression levels of the two human CARs were certainly not statistically superior, but the CARs with CD8a-HD (36 amino acids), which consists of more amino acids than CD28-HD (26 amino acids), were more highly expressed on human T cells. Although a more detailed study focusing on the HD length is needed in the future, it is well assumed that the HD length affects the CAR expression efficiency on the cell surface in human CARs.
Q17;
Line 344: the conclusion what the exact role of the TMD is could only be drawn if constructs were included that differed with respect to their TMD only. Otherwise, it is always possible that the HD also contributes to the effect.
A17;
Comparison of the expression patterns between mV/4/3z/3z and mV/4/4/3z, mV/8a/3z/3z and mV/8a/8a/3z, or mV/28/3z/3z and mV/28/28/3z in Figure 2 clearly shows that the surface expression stability and cellular localization of CAR differ depending on the structural differences between CD3z-TMD and each immune molecule-derived TMD. Therefore, it is reasonable to conclude that TMD regulates the CAR signaling intensity (amount) by modulating the CAR surface expression level.
Again, thank you for giving us the opportunity to strengthen our manuscript with your valuable comments and queries. We have worked hard to incorporate your feedback and hope that these revisions persuade you to accept our submission.

Reviewer 2 Report
Authors present a high-quality and very well-written experimental manuscript that describes how hinge and transmembrane domains of CAR regulate receptor expression and activation threshold.
Authors generated different chimeras and analyzed their expression levels and antigen-specific activity on CAR-T-cells. Studies hinge/transmembrane domains included the ones originated from CD4, CD8a and CD28.
Authors used a wide range of methods that include cloning, flow cytometry, immunoblotting, RT-qPCR, cytotoxicity assay, ELISA.
Authors demonstrated that:
- high positive correlation was observed between CAR expression level and antigen-specific proliferation of each CAR-T-cell type;
- mouse CAR [mV/28/28/3z]-T cells showed significantly higher cytokine-producing ability than other assessed combinations;
- human CAR [hV/h28/h28/h28-h3z]-T cells showed significantly higher cytotoxic activity than human CAR [hV/h8a/h8a/h28-h3z]-T cells, and its cytotoxicity per CAR expression level was about 2.5 times higher;
- overall, in both human and mouse CAR-T cells, domain modification affects the CAR surface expression level as well as signal input efficiency.
Based on these results they conclude that:
- CAR expression level and stability on T-cell were greatly affected by the transmembrane rather than the hinge domain;
- Antigen-specific functions of most CAR-T cells depended on their CAR expression levels;
- CARs with CD8a or CD28 derived hinge domain showed significant differences in CAR-T-cell function despite their equal CAR expression levels.
Comments:
- Please comment on the expected speed of CAR-T exhaustion for highly toxic variants with CD28 domains. Does higher toxicity result in shorter potential persistence in clinical settings?
- Authors are kindly requested to cite the following article (doi:10.3390/cancers12010125) that reviews using CAR-T cells against certain types of tumors.
Overall, the manuscript is highly valuable for the CAR-T scientific community.
Author Response
Reply to the Review Report (Reviewer 2):
We appreciate you taking the time to offer us your comments related to the paper. We are very grateful that you have a good understanding of the new findings on HD/TMD that we wanted to convey in this paper. Our responses are given in a point-by-point manner below. Changes in the manuscript are underlined and highlighted in yellow.
Q1;
Please comment on the expected speed of CAR-T exhaustion for highly toxic variants with CD28 domains. Does higher toxicity result in shorter potential persistence in clinical settings?
A1;
If CAR-T cells with high cytotoxic activity are rapidly eliminated in vivo, we suspect that this may be due to the induction of activated induced cell death of CAR-T cells rather than their exhaustion. Therefore, we believe that it is necessary to design CAR structures to regulate CAR-T cell function according to the localization of CAR target antigen and its intensity. For example, if the antigen expression level is high, such as CD19, and CAR-T cells are prone to contact with antigen-expressing cells, it may be better to design a CAR incorporating CD3z-TMD or CD4-TMD with low CAR expression stability and CD8a-HD with a high signaling threshold.
Q2;
Authors are kindly requested to cite the following article (doi:10.3390/cancers12010125) that reviews using CAR-T cells against certain types of tumors.
A2;
We cited your article [reference 20] along with the text below in the Discussion.
Page 11, lines 317-322:
“Although CAR-T cell therapy has proven to be a breakthrough therapy with potent and sustained antitumor activity, the occurrence of serious side effects and the low efficacy in solid tumors have gradually become apparent as a challenge. Therefore, many researchers are investigating the possibility of improving the function and persistence of CAR-T cells by modifying the CAR structure, developing a CAR-T cell generation protocol, and combining it with existing anti-cancer drugs [20-22].”
Again, thank you for giving us the opportunity to strengthen our manuscript with your valuable comments and queries. We have worked hard to incorporate your feedback and hope that these revisions persuade you to accept our submission.

Reviewer 3 Report
please see attached file

Author Response
Reply to the Review Report (Reviewer 3):
We appreciate you taking the time to offer us your comments and insights related to the paper. We found your feedback very constructive and tried to be responsive to your concerns. Our responses are given in a point-by-point manner below. Changes in the manuscript were clearly indicated with yellow high lights, along with the use of the "Track Changes" function in Microsoft Word.
Point 1;
The human CAR panel is quite small (only two receptors). Therefore, the authors should at least discuss the knowledge on other receptor formats and the influence of their hinge/transmembrane domains on expression and function published previously (e.g. Krug. et al., doi: 10.1007/s00262‐015‐1767‐4.)
Response 1;
Many existing CARs have CD8α- or CD28-derived HD/TMD, regardless of the type of their target antigens. This is assumed to be the result of the empirical selection of HD/TMD structures with superior CAR expression efficiency and cell surface expression stability, as we have shown in this study. In the development of second- and third-generation CARs, it has been reported that CAR signals can be input into the cell more efficiently by designing the TMD to be derived from the same molecules as the STD proximal to the cell membrane (Guedan S. et al., doi: 10.1172/jci.insight.96976.). As we have consistently noted in the Discussion section, the role of HD/TMD is less clear than that of ARD and STD. Therefore, we believe it is necessary to analyze the structural role of HD/TMD in detail, and to consider the combination of HD/TMD and STD in the future. These knowledges have been added to the Discussion section as follows.
Page 11-12, lines 329-330;
“In contrast, the HD/TMD of most CARs has empirically selected CD8α-, CD28-derived sequences as structures that can be efficiently expressed on T cells, and their role in CAR activity and CAR-T cell functions are poorly understood.”
Page 14, lines; 406-410;
“Furthermore, the analysis of the combination of HD/TMD and STD is also an important consideration to apply the structural information of HD/TMD to the second- and third-generation CARs for clinical use because the previous study has shown that TMD derived from the same molecules as STD proximal to the plasma membrane is more efficient for CAR signal input [38].”
The literature you presented described very interesting results in the development of mRNA-transfected CAR-T cells. However, it was difficult to determine whether CAR expression and activity were affected by HD/TMD or STD because the HD/TMD of the first-generation CAR and the second-generation CAR were different. On the contrary, it is a very important finding that differences in the scFv structure as the ARD affected the CAR expression profiles in both first- and second-generation CARs. Therefore, we have added the following text and its references to the text.
Page 13, lines 345-348;
“Recent studies have reported different expression levels of CAR depending on scFv structure even in the same HD/TMD/STD structure [14,32], and we have shown that the CAR surface expression efficiency is affected by differences in the structural stability of scFv [14].”
(Reference 32 is the literature of Krug. et al., doi: 10.1007/s00262‐015‐1767‐4.)
Point 2;
RNA electroporation allows for the exact control of CAR expression on the cell surface by adjusting the quantity of mRNA electroporated. Although the expression level of the two human CARs was not significantly different, the authors should try to get more equivalent expression by adjusting the mRNA quantities during electroporation and perform the cytotoxicity test with these new cells.
Response 2;
We agree that CAR mRNA electroporation is an excellent method to efficiently transfect T cells with CAR mRNA and to increase or decrease the CAR expression level by modulating the amount of its mRNA. In this study, we transfected T cells with equal amounts of CAR mRNA to correctly assess the effect of HD/TMD structure on the mechanism of CAR expression without biasing the amount of CAR protein synthesized. While both human CARs showed high CAR expression level, the VEGFR2-specific cytotoxic activity was significantly higher in the CAR with CD28-HD/TMD, which had a slightly lower CAR expression level. These results were analyzed very reproducibly in three separate trials, and their mean values are shown in Figure 4. We believe that these results, even with the current test content, have sufficient scientific basis to guide the reader to the conclusions of this paper, “CAR HD/TMD regulate CAR expression and signaling threshold”.
Although mRNA electroporation method can increase or decrease the CAR expression level by modulating the amount of CAR mRNA, it is very difficult to achieve complete agreement between the two CAR expression profiles. The generation of CAR-T cells with matched CAR expression level by modulating the amount of mRNA transfection may better define the results of the cytotoxic activity per CAR expression shown in Figure 4C, but it is unlikely to reveal new important findings beyond those already presented in this paper. Therefore, we believe that elucidating the structural role of HD/TMD and clarifying the linkage between HD/TMD and STD will contribute to future CAR research rather than examining the regulation of CAR mRNA transfection.
Again, thank you for giving us the opportunity to strengthen our manuscript with your valuable comments and queries. We have worked hard to incorporate your feedback and hope that these revisions persuade you to accept our submission.
